# Myalgic Encephalomyelitis/Chronic Fatigue Syndrome and Post-COVID Syndrome: A Common Neuroimmune Ground?

**DOI:** 10.3390/diagnostics13010066

**Published:** 2022-12-26

**Authors:** Varvara A. Ryabkova, Natalia Y. Gavrilova, Tamara V. Fedotkina, Leonid P. Churilov, Yehuda Shoenfeld

**Affiliations:** 1Laboratory of the Mosaic of Autoimmunity and Department of Pathology, Saint Petersburg State University, 199034 Saint-Petersburg, Russia; 2Department of Hospital Therapy Named after Academician M.V. Chernorutskii, Research Institute of Rheumatology and Allergology, Pavlov First Saint Petersburg State Medical University, 197022 Saint-Petersburg, Russia; 3National Medical Research Center Named after V. A. Almazov, 197341 Saint Petersburg, Russia; 4Saint Petersburg Research Institute of Phthisiopulmonology, 191036 Saint Petersburg, Russia; 5Zabludowicz Center for Autoimmune Diseases, Sheba Medical Center, Tel-Hashomer, Ramat-Gan 52621, Israel; 6Ariel University, Ariel 98603, Israel

**Keywords:** chronic fatigue syndrome, post-COVID syndrome, postural orthostatic tachycardia, microcirculation, immune system

## Abstract

Myalgic encephalomyelitis/chronic fatigue syndrome (ME/CFS) is a debilitating chronic disease of unknown etiology, sharing a similar clinical presentation with the increasingly recognized post-COVID syndrome. We performed the first cross-sectional study of ME/CFS in a community population in Russia. Then we described and compared some clinical and pathophysiological characteristics of ME/CFS and post-COVID syndrome as neuroimmune disorders. Of the cohort of 76 individuals who suggested themselves as suffering from ME/CFS, 56 were diagnosed with ME/CFS by clinicians according to ≥1 of the four most commonly used case definitions. Of the cohort of 14 individuals with post-COVID-19 syndrome, 14 met the diagnostic criteria for ME/CFS. The severity of anxiety/depressive symptoms did not correlate with the severity of fatigue either in ME/CFS or in post-COVID ME/CFS. Still, a positive correlation was found between the severity of fatigue and 20 other symptoms of ME/CFS related to the domains of “post-exertional exhaustion”, “immune dysfunction”, “sleep disturbances”, “dysfunction of the autonomic nervous system”, “neurological sensory/motor disorders” and “pain syndromes”. Immunological abnormalities were identified in 12/12 patients with ME/CFS according to the results of laboratory testing. The prevalence of postural orthostatic tachycardia assessed in the active orthostatic test amounted to 37.5% in ME/CFS and 75.0% in post-COVID ME/CFS (the latter was higher than in healthy controls, *p* = 0.02). There was a more pronounced increase in heart rate starting from the 6th minute of the test in post-COVID ME/CFS compared with the control group. Assessment of the functional characteristics of microcirculation by laser doppler flowmetry revealed obvious and very similar changes in ME/CFS and post-COVID ME/CFS compared to the healthy controls. The identified laser doppler flowmetry pattern corresponded to the hyperemic form of microcirculation disorders usually observed in acute inflammatory response or in case of systemic vasoconstriction failure.

## 1. Introduction

Myalgic encephalomyelitis/chronic fatigue syndrome (ME/CFS) is a chronic acquired disease characterized by pathological fatigue, which worsens after physical or mental exertion (so-called post-exertional malaise), unrefreshing sleep, cognitive impairment, pain syndromes, symptoms of dysautonomia, neuroendocrine, and immune dysfunctions [1]. The prevalence of ME/CFS, according to the recent systemic review, reaches 0.89%, and the disorder is characterized by an approximately 1.5-fold predominance of women [2]. A Norwegian population-based study revealed two age peaks for the incidence rate: ages 10–19 and 30–39 years [3]. It was noted that in some countries (e.g., Poland), ME/CFS is diagnosed very rarely, which could be related to a lack of knowledge and understanding of ME/CFS among general practitioners and the lack of detailed and uniform clinical care guidelines [4,5]. To date, the cases of ME/CFS in Russia have neither been reported in the English-language medical literature nor have their prevalence estimated. There are currently no ME/CFS clinical services in Russia. Given the global prevalence of ME/CFS, it could be suggested that more than 2.5 million Russian citizens may suffer from ME/CFS remaining undiagnosed.

ME/CFS is a topic of growing interest now in view of the post-COVID syndrome as an increasingly recognized new clinical entity [6,7,8,9,10]. Early studies which described a constellation of symptoms experienced by 10–30% of people after the resolution of acute COVID-19 suggested many overlaps between the clinical presentation of the post-COVID-19 syndrome and ME/CFS [7,8]. Emerging longitudinal studies tracking post-COVID-19 patients’ symptoms over 6 months or more (as this is one of the criteria for ME/CFS) confirm these observations [8]

While the pathogenesis of ME/CFS and post-COVID-19 syndrome remain poorly understood, it seems that the underlying pathology involves the CNS; the autonomic nervous system; and a persistent, dysregulated immune and metabolic response to the infectious agents [10]. Previous studies have shown a high propensity of patients with ME/CFS to be misdiagnosed with a psychiatric condition [11]. At the same time, adults with ME/CFS frequently experience co-morbid depression and/or anxiety. Whether these mental health comorbidities relate to the severity of fatigue remains unknown.

We hypothesized that ME/CFS and post-COVID syndrome are not primary mental health conditions but share some clinical features and common pathophysiological mechanisms related to autonomic dysfunction and microcirculation disorder.

## 2. Materials and Methods

### 2.1. Participants

We performed the first pilot cross-sectional study of ME/CFS in a community population in Russia. Then we described and compared some clinical and pathophysiological characteristics of ME/CFS and post-COVID syndrome as neuroimmune disorders. Following approval from The Ethics Committee of St Petersburg University, individuals with ME/CFS and post-COVID syndrome as well as healthy volunteers, were recruited as part of a study of autoimmune dysautonomia. Requests for participation were made through messages to online support groups and social media. Post-COVID syndrome was defined as the presence of symptoms and/or signs of damage to various organ systems that develop during or after a previous infection with COVID-19, persisting for more than 12 weeks, and cannot be explained by an alternative diagnosis [12].

Three cohorts were formed:A cohort of patients from 18 to 75 years old who met ≥1 of the 4 most commonly used ME/CFS case definitions (the Fukuda et al. (1994) CFS criteria [13], the Canadian ME/CFS criteria [14], the Myalgic Encephalomyelitis International Consensus Criteria (ME-ICC) [15], and the Institute of Medicine criteria), in whom the onset of the disease was not associated with COVID-19;A cohort of patients from 18 to 75 years old who met ≥1 of the 4 most commonly used ME/CFS case definitions and those symptoms developed following COVID-19;Control group (healthy volunteers from 18 to 75 years old).

The exclusion criteria were:

For the first group: the presence of any of the diseases/conditions from Table 1 unless complete remission was achieved.

For the second cohort:The presence of any symptoms (including chronic fatigue) before COVID-19.

For the third group:Complaints of chronic fatigueIn cases of previous viral infections, including COVID-19 <4 weeks after recovery to the moment of enrollment in the study.

### 2.2. Symptom Assessment Tools

#### The ME/CFS Symptom Questionnaire DePaul Symptoms Qustionnaire-2

(DSQ-2) is a revised version of DSQ-1, a standardized self-report questionnaire that assesses the symptomatology of ME/CFS, as well as medical, psychiatric, and social history data [17]. DSQ has demonstrated high reliability and validity and the ability to accurately differentiate ME/CFS patients with other chronic diseases and healthy people from the control group [18]. In DSQ-2, participants rated the frequency and severity of each symptom over the past 6 months on a five-point Likert scale (frequency scale: 0 = never during this time; 1 = rarely (up to 1–2 times a week); 2 = often (3–4 times a week); 3 = very often (almost every day); 4 = every day; severity scale: 0 = no such symptom; 1 = mildly disturbing symptom; 2 = moderately disturbing symptom; 3 = symptom quite disturbed; 4 = very disturbed symptom). Further processing of the results was carried out by correlating the answers of the subject with a key that allows determining the participant’s compliance with the four most common sets of diagnostic criteria for ME/CFS in the world clinical and research practice: Fukuda et al. (1994); Canadian ME/CFS (Carruthers et al., 2003); ME-ICC (Carruthers et al., 2011); Institute of Medicine (IOM, 2015). Composite scores were also calculated for each symptom by averaging the scores for the frequency and severity of the symptom and multiplying by 25 to obtain scores from 0 to 100 (higher scores indicate more pronounced symptom manifestation) [18].

The Hospital Anxiety and Depression Scale (HADS) [19] was used for the assessment of anxiety and depression in this study. It has been shown that HADS is a reliable scale for identifying and assessing the severity of symptoms of anxiety and depression, both among patients with somatic diseases and among patients with mental disorders, both in primary care patients and in the general population [20]. The participants filled in the questionnaire by themselves. The score of 0–7 in each subscale represents “normal”; 8–10 “doubtful case of anxiety/depression”; 11—“probable case of anxiety/depression”.

### 2.3. Patient Recruitment

After filling out the questionnaire, clinicians assessed the patients and diagnosed them with ME/CFS according to the international standard [21]. In order to exclude diseases indicated as the exclusion criteria for ME/CFS, the history of the present disease and personal medical history were taken. The results of the laboratory tests, instrumental evaluation, and medical reports from the specialists were analyzed. The results of the immunological testing were analyzed separately for those patients who got tested before.

### 2.4. Active Orthostatic Test

We performed the active orthostatic test according to the protocol of F. Schellong [22]. The participants were informed about the need to refrain from consuming caffeine- and alcohol-containing beverages and smoking on the day of the test. The last meal was to be light, no later than 3 h before the test. These recommendations were made in order to minimize the influence of external factors on hemodynamic parameters. We performed the test no later than 4 p.m. as orthostatic intolerance is more pronounced in the morning. During the test, we placed a compression cuff for measuring blood pressure on the shoulder of the participant and did not remove it until the end of the study. We also placed a pulse oximeter on the index finger of the other hand to continuously measure heart rate. The participant was asked to lie quietly on the couch for 10 min. After that, we measured blood pressure and heart rate. We took these values of blood pressure and heart rate as the baseline, and the patient was asked to stand up calmly, spread his legs shoulder-width apart, and stand relaxed for 10 min. Immediately after getting up, the heart rate was determined, and then the blood pressure and heart rate were measured, and the person’s subjective feelings were assessed at the end of each subsequent minute. POTS was diagnosed with an increase in heart rate by 30 or more per 1 min in the standing position and the absence of orthostatic hypotension (drop in systolic blood pressure >20 mm Hg) [23]. This increase in heart rate had to be stable—that is, to manifest at least in two consecutive heart rate measurements [23].

### 2.5. Assessment of Microcirculation

We assessed microcirculation by amplitude–frequency wavelet analysis of blood flow oscillations with laser Doppler flowmetry (LDF) [24]. Because of the proven parallelism between microcirculatory changes in the skin and inner organs, we investigated forearm blood flow in the participants with «LASMA MC-1» peripheral blood and lymph flow laser diagnostic complex (LASMA LLC, Russia). Participants were examined once, with the diagnostic probe placed on the external surface of the right forearm for 2 min. The protocol for the study of microcirculation with LDF included: 1. Determination of the average value of tissue perfusion with blood, M; 2. Determination of the mean square deviation of M oscillations in a given time interval, σ; 3. Determination of the oscillation index, IFM. 4. Spectral analysis of biorhythms of tissue blood flow oscillations with the determination of oscillation amplitudes in given frequency ranges: low frequency (LF) 0.05–0.2 Hz, high frequency (HF) 0.2–0.4 Hz, pulse frequency (PF) 0.8–1.6 Hz, as well as determining the contribution of individual frequency ranges to the total power of the spectrum of biorhythms. 5. Determination of microvascular tone and vascular resistance. Amplitude-frequency wavelet analysis of blood flow oscillations was performed. Time-averaged vasomotion amplitude was assessed using maximum values (Amax) in the corresponding frequency band (Fmax). The whole technique was detailed elsewhere [25]. The following indices were calculated: the contribution of slow oscillations (vALF) (0.05–0.2 Hz); the contribution of fast (venular) oscillations (vAHF) (0.2–0.4 Hz); contribution of pulse oscillations (vACF) (0.8–1.6 Hz); IFM, vascular resistance and microvascular tone. The contribution of the corresponding frequency range (v: vALF, vAHF, vACF) was determined as the percentage of the squared amplitude of a given range (A) to the total spectrum power (M), which is the sum of the squared amplitudes over 3 ranges.
M = A^2^LF + A^2^HF + A^2^CF(1)
V = A^2^/M × 100%(2)

The oscillation index (IFM) is an indicator of the ratio of the mechanisms of active and passive modulation of tissue blood flow and is determined by the ratio of the average amplitudes of oscillations:IFM = ALF/(AHF + ACF)(3)

This index characterizes the overall efficiency of microcirculation regulation. Vascular resistance was calculated as the ratio of the sum of the amplitudes of fast and pulse oscillations and the mean square deviation of M:R = (AHF + ACF)/σ(4)

Normalization of the amplitude of low-frequency oscillations (ALF) relative to the mean square deviation of M allows us to assess the microvascular tone, which was calculated as:CT = σ/ALF(5)

Perfusion value (M), standard deviation (σ), and amplitudes of blood flow modulation sections were assessed in arbitrary perfusion units.

### 2.6. Statistical Analyses

Statistical analyses were conducted with GraphPad Prism v. 9.1.1. (GraphPad Software, San Diego, CA, USA) with the assumed level of statistical significance of α < 0.05. The data normality was assessed using the Shapiro–Wilk test and visual histogram inspection. Statistical characteristics of measured values were presented for the normal distribution as arithmetic means ± SD and for non-normal distribution as the median and interquartile range [25%; 75%]. Depending on the number of independent samples, the Mann–Whitney U test or the Kruskal–Wallis were used to evaluate the significance of differences between measured values obtained in the cohorts. When results were significant, they were further explored using the post-hoc Dunn’s test. The effect size was calculated using the effect size r for the Mann–Whitney U test. Interpretation of r is as follows: r = <0.3 (small effect size), r = 0.3–0.5 (moderate effect size), r ≥ 0.5 (large effect size). For categorical variables, the Pearson χ-square test and Fisher’s exact test were performed. For the χ-square test, OR was calculated as an effect size indicator. Pearson correlation tests were used to assess the association between the composite score for “Fatigue/Extreme tiredness” (question 13 in DSQ-2) and: (1) composite score for every other of the 90 symptoms from the DePaul Symptom Questionnaire; (2) HADS-A and HADS-D subscales scores.

## 3. Results

### 3.1. Participants

Among the members of the community of patients who suspected they suffered from s ME/CFS, 76 people completed the questionnaire DSQ-2. Of these, 56 people who met all inclusion and exclusion criteria were diagnosed by clinicians with ME/CFS and selected for the first cohort. In total, five people were excluded from the study because they did not meet any of the four ME/CFS clinical definitions. In total, 15 people were excluded from the study because they met the exclusion criteria (i.e., suffered from other medical conditions that could potentially cause chronic fatigue). Among the first cohort, 52 participants met the Fukuda criteria, 50 met the IOM criteria, 38 met the Canadian ME/CFS criteria, and 32 met the ME-ICC. The overlap between subgroups is shown in Figure 1A.

Among the members of the community of patients with the post-COVID syndrome, 15 people completed the DSQ-2 questionnaire. Of these, 14 people were included in the second cohort. Then, one person was excluded because he was over 75 years of age. Among the second cohort, 13 participants met the Fukuda criteria, 13 met the IOM criteria, 8 met the Canadian ME/CFS criteria, and 8 met the ME-ICC. The overlap between subgroups is shown in Figure 1B.

The control group of healthy individuals consisted of nine volunteers who met all the inclusion and exclusion criteria.

The demographics of the sample population are summarized in Table 2. Participants from the three groups were comparable regarding their age and sex.

### 3.2. Clinical and Anamnestic Characteristics of ME/CFS as Neuroimmune Disorder

We obtained information about possible triggers for the development of ME/CFS from the answers of the participants of the 1st cohort to the 110th question of DSQ-2 (“Did your fatigue/energy related illness start after you experienced any of the following? Check one or more and please specify”). In total, 35 people (62.5%) chose the response option “an infectious illness”, and 23 people (41.1%) chose the response option “severe stress(s) (bad or unhappy event(s)).” At the same time, both these answer options were chosen by 13 people (23.2%).

Analysis of the relationship between the composite score of fatigue and every other of the 90 symptoms from DSQ-2 revealed a statistically significant positive correlation between the severity of fatigue and the other 20 symptoms listed in Table 3.

Seven of these symptoms belonged to the “post-expression malaise” domain, which is one of the key manifestations of ME/CFS [1]. In total, four symptoms belonged to the domain of “immune dysfunction”, four symptoms—to the domain of “sleep disorders”, two symptoms to the domain of “neurological sensory/motor disorders”, two symptoms to the domain of “dysfunction of the autonomic nervous system”, and one to the domain of “pain syndromes”. The data obtained were consistent with the concept of ME/CFS as a neuroimmune disease.

### 3.3. Mental Health Screening in Patients with ME/CFS and Assessment of the Relationship between Mental Health and Severity of Fatigue

The HADS questionnaire was completed by 46 patients from the first group, 14 people from the second group, and 9 people from the third group. As can be seen from Table 4, the median values of the severity of depression according to HADS-D among the studied groups are higher in the groups of patients compared with healthy controls.

At the same time, there was a statistically significant difference between the median values of the severity of anxiety according to HADS only between the first cohort and the healthy controls (Table 5).

In order to test the hypothesis about the relationship between fatigue and mental health in the first group, a correlation analysis was performed. There was no significant relationship between the severity of the depressive or anxiety symptoms and the severity of fatigue (Table 6).

### 3.4. Assessment of the Immune Status in Patients with ME/CFS

In total, 12 patients from the 1st cohort had previous primary immunological examination results, which consisted of a comprehensive assessment of the lymphocyte subpopulations. In some cases, it was supplemented by an assessment of the level of circulating immune complexes, C3-, C4-components of complement, the study of interferon status, and indicators of the function of the granulocytes and monocytes. Even though 5/12 patients had an increase in the relative number of CD3+ cells, an increase in the absolute number of these cells was found only in one patient. The same patient had an increase in the number of CD3+ CD4+ cells. Another patient had a decrease in the number of CD3+ CD4+. Despite a decrease in the absolute number of CD3+ CD8+ cells detected in the single patient, five patients had an increase in the immunoregulatory index (CD4+/CD8+) of more than 2.0. At the same time, two patients had a decrease in the immunoregulatory index of less than 1.0. In five people, laboratory analysis included determining the absolute content of double-positive CD4+ CD8+ cells. In 3/5 patients, these cells were lower than the 25% percentile of the reference values. The relative number of NK cells, determined in all 12 patients, was reduced in three patients according to the latest data on reference values in the Russian population [26]. Moreover, one patient had an increase in the number of these cells. However, in 8/12 patients number of NK cells was lower than 25% of the reference values. Most of the patients had normal B lymphocyte counts. Meanwhile, one person showed a decrease, and one showed an increase in the number of these cells. An additional analysis of subpopulations of B lymphocytes carried out in these two people revealed that in the patient with an increase in the total number of B lymphocytes, it occurred mainly due to B2 lymphocytes and B memory cells. The level of the C3 component of complement was determined in five people—in two of them, an increase in its level was recorded, and in one person—a decrease. The serum levels of interferons (IFNs) and basal and induced levels of the IFNs secretion by leukocytes were analyzed in four people. All of them had an increase in serum IFNα levels. Moreover, the induced production of IFNα by leukocytes was reduced in 3/4 of the people. The induced secretion of IL-1b, measured in three people, was markedly increased in all patients. Assessment of the functional properties of granulocytes and monocytes was carried out in three people. However, due to the difference in the methods used (inhibition of leukocyte migration, Nitroblue Tetrazolium Test, killing coefficient, phagocytic index, phagocytic number), it was not possible to compare the data between the participants.

### 3.5. Identification of Autonomic Dysfunction during an Active Orthostatic Test

An active orthostatic test was performed on some participants from all three cohorts. In total, 6/16 people in cohort 1, 6/8 people in cohort 2, and 1/6 healthy controls met the criteria for postural orthostatic tachycardia syndrome (POTS). As can be seen from Table 7, POTS was statistically significantly more frequent in the ME/CFS group that developed after COVID-19 than in the control group of healthy individuals.

An alternative form of orthostatic intolerance, i.e., orthostatic hypotension, was detected only in one person in cohort 1, one person in cohort 3, and none of the people in cohort 2.

To test the hypothesis that the hemodynamic disorder of the POTS type is one of the characteristics of the ME/CFS that developed after COVID-19 and that there exists a causal association between these two conditions rather than a coincidence, we calculated the average increase in heart rate relative to the basal values in all three study cohorts for each minute of the test. Pairwise comparison of these values between cohorts at each minute of the test revealed that the ME/CFS developing after COVID-19 was characterized by a statistically more pronounced increase in heart rate at the 6th, 7th, 8th, 9th, and 10th minute of the test compared with the control group and at the 8th and 9th minutes of the test compared with the ME/CFS developed outside the context of COVID-19. These results support our hypothesis.

### 3.6. Assessment of the Dynamic Characteristics of Microcirculation with Laser Doppler Flowmetry

A non-invasive study of blood microcirculation parameters using the LDF method was carried out on 10 participants from cohort 1, 7 participants from cohort 2, and 7 healthy individuals. The parameters of skin microcirculation in the cohorts are presented in Table 8 and explained in Figure 2.

As follows from Table 8, there was a statistically significant increase in the average value of tissue perfusion with blood and a decrease in IFM in participants with ME/CFS, including ME/CFS developed following COVID-19, compared with the control group. There was a statistically significant decrease in the contribution of low-frequency oscillations (0.05–0.2 Hz) to the total power of the spectrum of biorhythms detected in the cohort of ME/CFS that developed after COVID-19. In the case of ME/CFS of a different origin, this decrease was at the border of statistical significance. At the same time, statistically significant differences between cohorts 1 and 2 were revealed only in terms of vascular resistance, which was higher in the first cohort.

## 4. Discussion

In this study, for the first time in the Russian population, a validated tool (DSQ-2) was used to identify those cases meeting the four most common ME and CFS case definitions. We tested members of the Internet community of patients who suspected they had ME/CFS and members of the Internet community of patients with symptoms persisting for more than 12 weeks after COVID-19. The results of filling out the questionnaire showed the validity of the assumption of most members of the Internet community of patients who suspected they had ME/CFS about their disease—56/76 people (73.7%) met at least one of the case definitions for ME/CFS and were diagnosed with ME/CFS by a clinician. In total, 15 people who met the inclusion criteria according to the DSQ-2 questionnaire were excluded from the study because they met the exclusion criterion (the presence of other diseases that can potentially cause chronic fatigue). This data indicated the key role of a thorough examination of people with complaints of chronic fatigue in clinical practice for the purpose of differential diagnosis. The median age of patients in the first group was 39.3 [31.4; 45.9] years old, and the male/female ratio was 2:1, which corresponds to the data of epidemiological studies conducted in other countries where ME/CFS is more recognized [2].

In 2021, against the backdrop of the COVID-19 pandemic, the ME/CFS problem acquired additional urgency. The fact is that at least some of the patients after the acute phase of COVID-19 (often experienced just in a mild form) acquired some prolonged symptoms. Among them, fatigue was the most common [27]. Despite negative nasal-swab PCR, these individuals continue to suffer from various symptoms that persist for more than 12 weeks after the onset of the disease or appear after the acute phase of COVID-19. An increasing number of publications point to a pronounced similarity between the clinical manifestations of the post-COVID syndrome and ME/CFS. Notably, ME/CFS in 70% of cases also develops following an infectious disease [28].

In order to test the hypothesis of the relationship between ME/CFS and post-COVID syndrome, the DSQ-2 questionnaire was sent to the members of the Internet community of patients with symptoms persisting for more than 12 weeks after COVID-19. The fact that all individuals who completed the questionnaire met at least one of the ME/CFS case definitions confirms the presence of a close relationship between post-COVID syndrome and ME/CFS.

In the first group of patients (in whom the development of ME/CFS was not associated with COVID-19), the infectious disease was indicated as a probable trigger for the development of ME/CFS by 62.5% of patients, which is similar to the data of other authors (70%) [28] and indirectly indicates the involvement of the immune system in the pathogenesis of the disease.

The key symptom of ME/CFS is severe fatigue that does not go away even after adequate rest and persists for more than 6 months. Some information about the possible mechanisms of chronic fatigue in ME/CFS can be obtained from the correlation analysis between the severity of this symptom and the presence and severity of other symptoms of ME/CFS, divided into several domains, remembering, however, that the identification of a positive correlation does not yet indicate the presence causal relationship. The statistically significant positive association between fatigue that does not go away after adequate rest and 20 other symptoms of ME/CFS allows us to draw the following conclusions. Firstly, the fact that 7 out of 20 symptoms belong to the domain of “post-expression malaise” additionally indicates the key role of this phenomenon in ME/CFS. There is evidence that the two-day cardiopulmonary exercise test objectively measuring PEM makes it possible to distinguish ME/CFS patients from healthy individuals [29], which is of great importance in the absence of reliable laboratory biomarkers of ME/CFS. Deterioration of the patient’s condition after physical activity should serve as a warning for medical doctors against recommending ME/CFS patients increase the level of physical activity without careful supervision by a rehabilitation specialist. Secondly, the fact that 4 out of 20 symptoms belong to the domain of “immune dysfunction” (this domain contains seven questions in total), despite the non-obviousness of the pathophysiological connection of pathological fatigue, for example, with soreness of the lymph nodes or symptoms of sinusitis, indicates the important role of immune dysfunction in the development of symptoms of the disease and justify the usage of the immunomodulatory drugs to alleviate the main symptom (fatigue). Thirdly, the belonging of four symptoms to the domain of “sleep disturbance” and one symptom to the domain of “pain syndromes” confirms the validity of the treatment approach to ME/CFS, described in the recent guidelines from the European Network on Myalgic Encephalomyelitis/Chronic Fatigue Syndrome (EUROMENE) [30]. According to this approach, to reduce the severity of fatigue, patients are provided first of all with symptomatic help to normalize sleep and combat pain. Finally, the belonging of four symptoms to the domains associated with disorders of the nervous system (autonomic, sensory, and motor functions) confirms the classification of ME/CFS in ICD-10, where it belongs to the chapter G—“Diseases of the nervous system”.

Our data, therefore, not only agree with the concept of ME/CFS as a disease with neuroimmune pathogenesis but also allow us to make assumptions about approaches to diagnosis and treatment of this disease, as well as the most effective organization the patient care in ME/CFS, which optimally should be provided by neurologists.

In the past, ME/CFS has often been misdiagnosed as a psychiatric disorder of the affective spectrum, leading to mismanagement and deterioration of the patient’s health [31]. Today, it is believed that anxiety and depressive symptoms, which are common in ME/CFS [32,33], should not always be considered as a sign of an alternative diagnosis. The prevalence of clinically significant and subclinical anxiety and depression in ME/CFS, determined based on the Hospital Anxiety and Depression Scale (HADS) in our study, is consistent with the literature [34]. At the same time, it should be remembered that recent research data on this topic suggests that depression and anxiety in ME/CFS are associated with the neuroinflammation process, pain syndromes, psychological distress due to the inability to return to work, and reduced physical functioning, social isolation, as well as insufficient knowledge of medical specialists about the disease. Consequently, they are deontologically vulnerable and have skeptical attitudes toward the patient’s problems [31]. In addition, the absence of a statistically significant difference between the prevalence of anxiety and depressive symptoms between the group of patients with ME/CFS and healthy individuals, as well as the lack of correlation between the severity of fatigue and psychological distress (anxiety and depression) in the group of patients with ME/CFS, calls to question the potential effectiveness of the treatment of depressive and anxiety symptoms in order to reduce the severity of fatigue. This is consistent with our experience with patients suffering from ME/CFS: just very few of them reported a positive effect of antidepressants on the severity of their main complaint—fatigue.

Deviations from the reference values for several parameters of the screening immunologic tests were observed in 12/12 patients (100%) who underwent this laboratory evaluation. The changes concerned lymphocyte subset profile, level of circulating immune complexes, C3 and C4 components of the peripheral blood complement, interferons levels and secretion, and indicators of the function of the granulocytes and monocytes. However, the clinical heterogeneity and polyetiological nature of ME/CFS, as well as the rapid dynamics of the indicators of the immunological status, were the reasons for the difference in the patterns of immune dysfunction in patients. A follow-up study should be performed to assess the relationship between the CD4+/CD8+ ratio and the disease’s clinical features, course, and outcome. In our study, we noticed that according to this ratio, patients slipped into three almost equal groups (decrease in CD4+/CD8+ <1.0, increase in CD4+/CD8+> 2.5, normal value). Analysis of small cell populations such as double-positive T-lymphocytes, which are rarely performed in routine clinical practice, may also be of interest for further research. A decrease in the absolute count of NK cells is often mentioned in the literature as an important sign of ME/CFS. Contrary to expectations, the absolute count of NK cells was reduced (according to the latest data on reference values in the adult Russian population) only in 3/12 patients. However, it is important to acknowledge that the normal level of NK cells does not exclude their functional insufficiency. Even though phagocyte activity was reduced in all (3/3) patients who underwent corresponding laboratory evaluation, the unification of research methods in this area is needed to compare the results obtained in different laboratories.

Several DSQ-2 questions are related to orthostatic intolerance because it is a frequent finding in ME/CFS [4]. In particular, 11 to 50% of patients with ME/CFS may suffer from postural orthostatic tachycardia syndrome (POTS). Interestingly, POTS itself often develops following infectious diseases and, at least in some cases, may be associated with the production of autoantibodies against adrenergic and cholinergic receptors [35]. We found that according to the results of the active orthostatic test, 37.5% of patients with ME/CFS met the diagnostic criteria for POTS. The diagnosis of POTS in this group is of clinical value because pharmacological and non-pharmacological methods allow one to cope with the symptoms of this condition, which can additionally affect the quality of life of patients with ME/CFS. However, even more interesting is the extremely high prevalence of POTS among patients with the post-COVID syndrome (6/8 (75.0%) versus 1/9 (11.1%) in the control group, *p* = 0.02). It is important to note that among the control group, 6/9 people (66.7%) were infected with COVID-19 and recovered >4 weeks ago at the time of enrollment in the study; however, POTS was detected in only 1 of them. Pairwise comparison of the average increase in heart rate relative to basal values between the cohorts at each minute of the active standing test showed that in ME/CFS that developed after COVID-19, there is a more pronounced increase in heart rate starting from the 6th minute of the test compared to the control group. This allowed us to assume that POTS is one of the key characteristics of ME/CFS of post-COVID genesis. It is important to note that in 4/13 people who met POTS criteria in our study, the required increase in heart rate was achieved only at the 8–10th minute of the active orthostatic test. This confirms the practice of carrying out the test in its complete (within 10 min) and not abridged (5 min) version.

The rhythmic characteristics of oscillatory processes in the microcirculation system are useful for the diagnosis of many diseases related to the changes in the microcirculation [36]. The LDF method allows a non-invasive assessment of human blood microcirculation system disturbances. In this work, LDF was applied to assess the dynamic characteristics of microcirculation in ME/CFS, including post-COVID genesis (in the latter subgroup—it was done for the first time, to the best of our knowledge). A change in the microcirculation index (increase or decrease) characterizes, respectively, an increase or decrease in perfusion. Its increase can be associated with a lower tonus of the arterioles, which leads to an arterial hyperemia, or with the congestion of blood in the venules and venous hyperemia. Regarding the regulation of microcirculation, there are “active” and “passive” mechanisms. The “passive” mechanisms include external factors that act outside the microcirculatory bed: a pulse wave and the suction action of the “respiratory pump” from the veins. “Active” factors directly affect the vessels of the microvasculature by periodically changing the resistance of blood vessels to blood flow through vasomotions and creating transverse fluctuations in blood flow. These active factors are sympathetic nerve fibers, smooth muscle cells of the vascular wall, and endothelium-derived regulatory molecules. When carrying out spectral analysis, the active factors correspond to low-frequency oscillations [37]. There are several forms of microcirculation disorders: arterial hyperemia, venous hyperemia, combined hyperemia, ischemia, and stasis [38]. The changes identified in this study in ME/CFS, including ME/CFS of post-COVID-19 nature, correspond to the hyperemic form of microcirculation disorders, which is characterized by increased blood flow into the microcirculatory bed. It is distinguished by a significant increase in the number of functioning capillaries, an increase in tortuosity, vasodilation, and an increase in the permeability of the vascular wall. This form of microcirculation disorder is usually observed in acute inflammatory response or other conditions of decreased systemic vasoconstriction [25] (Figure 3).

The state of microcirculation in ME/CFS, which developed before the COVID-19 pandemic, and ME/CFS of post-COVID-19 nature, differed only in terms of vascular resistance. Bond et al. showed that chronic oxidative stress could contribute significantly to the development of ME/CFS symptoms due to the development of endothelial dysfunction [39]. The relationship between chronic inflammatory processes and increased arterial stiffness is well-known [40]. An increase in vascular resistance in cohort 1 compared to cohort 2 may reflect the contribution of the chronic inflammatory process of a long course to microcirculation disorders and suggests the existence of long-term consequences of ME/CFS, in particular, an increased risk of cardiovascular diseases.

Figure 4 represents the clusters of symptoms that were correlated with fatigue in ME/CFS in our study and indicates some potential pathophysiological mechanisms which could contribute to the development of ME/CFS and post-COVID syndrome. The latter, in our opinion, is similar in many respects to post-infectious ME/CFS cases reported back in the 20th century. At the same time, the hypotheses which are reflected in the Figures need to be confirmed in further studies.

## 5. Conclusions

1. Among patients with symptoms persisting for more than 12 weeks after recovery from acute COVID-19, 100% of individuals met the diagnostic criteria for ME/CFS, which confirms the presence of a close relationship between post-COVID syndrome and ME/CFS.

2. We found a statistically significant positive relationship between fatigue that does not get better with adequate rest and 20 other symptoms of ME/CFS related to the domains of “post-exertional exhaustion” (7 symptoms), “immune dysfunction” (4 symptoms), “sleep disturbances” (4 symptoms), “dysfunction of the autonomic nervous system” (2 symptoms), “neurological sensory/motor disorders” (2 symptoms), and “pain syndromes” (1 symptom). These data not only agree with the concept of ME/CFS as a disease with neuroimmune pathogenesis but also allow us to make assumptions about the approaches to the diagnosis, treatment, and organization of care for patients with ME/CFS.

3. There was no correlation between anxiety/depressive symptoms and the severity of fatigue in ME/CFS. This can indirectly show that fatigue in ME/CFS is not a consequence of primary mental disorders.

4. Immune dysfunction was detected in 12/12 patients with ME/CFS (100%) based on the analysis of the results of the laboratory screening immunological evaluation.

5. The prevalence of POTS in patients with ME/CFS, especially with ME/CFS of the post-COVID-19 nature, is high. Still, POTS in this group of patients can be difficult to diagnose due to its delayed occurrence in the active orthostatic test.

6. Changes in microcirculation in ME/CFS (including ME/CFS of the post-COVID-19 nature) identified with the LDF method correspond to the hyperemic form of microcirculation disorders which is generally observed in acute inflammatory response or in case of the systemic vasoconstriction failure. It seems that increased vascular resistance may occur later in the disease course due to the chronic inflammatory process.

## Figures and Tables

**Figure 1 diagnostics-13-00066-f001:**
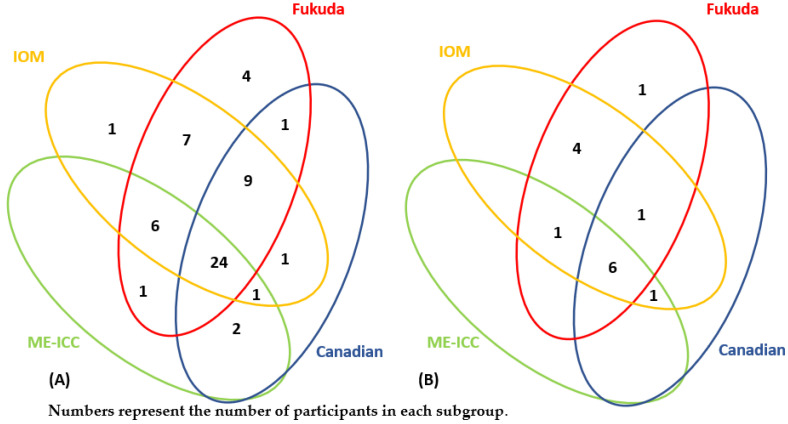
(**A**) The overlap between the subgroups that met different diagnostic criteria of ME/CFS in the first cohort (ME/CFS). (**B**) The overlap between the subgroups that met different diagnostic criteria of ME/CFS in the second cohort (post-COVID ME/CFS). Numbers represent the number of participants in each subgroup.

**Figure 2 diagnostics-13-00066-f002:**
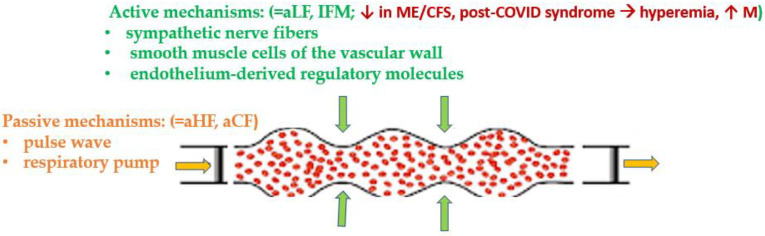
Interpretation of the microcirculation changes in ME/CFS and post-COVID ME/CFS. Active and passive mechanisms regulate microcirculation, as discussed below. The decreased activity of the active mechanisms underlies the hyperemic form of microcirculation disorder observed in ME/CFS and post-COVID-19 ME/CFS. aCF—contribution of pulse oscillation; aHF—contribution of high-frequency oscillation; aLF—contribution of low-frequency oscillation; IFM—oscillation index, M—average value of tissue perfusion with blood.

**Figure 3 diagnostics-13-00066-f003:**
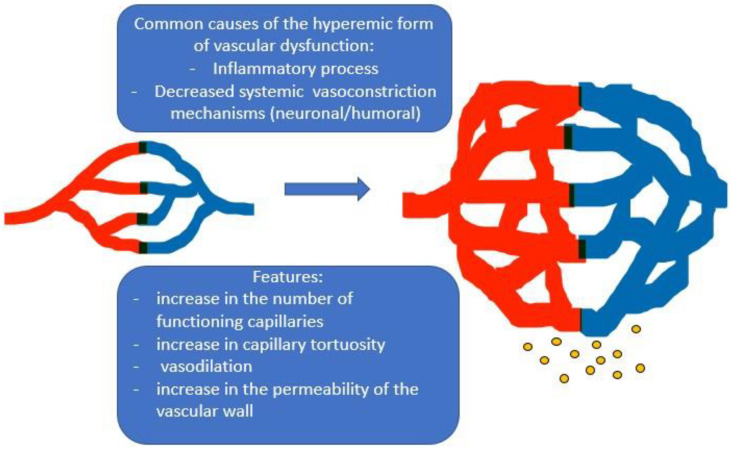
Features and common causes of the hyperemic form of microvascular dysfunction.

**Figure 4 diagnostics-13-00066-f004:**
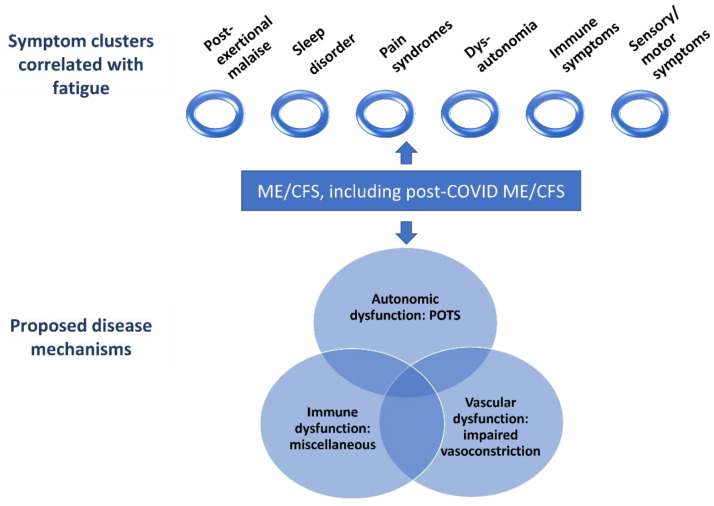
Visual summary of the main findings of the article. ME/CFS—myalgic encephalomyelitis/chronic fatigue syndrome, POTS—postural orthostatic tachycardia syndrome.

**Table 1 diagnostics-13-00066-t001:** Exclusion criteria for the first cohort (exclusion criteria for diagnosis of ME/CFS) [16].

-Endocrine diseases/metabolic disorders: primary adrenal cortex insufficiency, Cushing’s syndrome, hyper- and hypothyroidism, diabetes mellitus, hypercalcemia;
-Rheumatological diseases: systemic lupus erythematosus, rheumatoid arthritis, polymyositis;
-Hematological diseases: iron deficiency anemia, hemochromatosis, idiopathic thrombocytopenic purpura;
-Infectious diseases: HIV infection, hepatitis B, hepatitis C, tuberculosis, Lyme disease, giardiasis, helminthiasis, syphilis;
-Neurological diseases: multiple sclerosis, narcolepsy, obstructive sleep apnea, restless legs syndrome, Parkinson’s disease, myasthenia gravis, vitamin B12 deficiency, cervical spine injuries, epilepsy;
-Psychiatric illnesses: bipolar disorder, substance dependence, generalized anxiety disorder, schizophrenia, major depressive disorder;
-Gastrointestinal diseases: celiac disease, Crohn’s disease, ulcerative colitis
-Cardiovascular diseases with congestive heart failure;
-Chronic intoxication with heavy metals (lead, mercury);
-The development of the patient’s symptoms as side effects of any drugs;
-Respiratory diseases (chronic obstructive pulmonary disease, bronchial asthma) with the development of chronic respiratory failure;
-Overwork (work more than 50 h a week), overtraining syndrome;
-Body mass index over 40;

**Table 2 diagnostics-13-00066-t002:** Characteristics of the participants in ME/CFS, post-COVID ME/CFS, and controls cohorts.

Parameters	1st Cohort (ME/CFS)	2nd Cohort (Post-COVID ME/CFS)	3rd Cohort (Healthy Controls)	*p* Value
Number of participants	*n* = 56	*n* = 14	*n* = 9	
Age, median [interquartile range]	39.3 [31.4; 45.9]	34.9 [29.8; 40.2]	31.7 [22.2; 45.1]	0.28
Sex, male/female.	18/38	4/10	4/5	0.71

**Table 3 diagnostics-13-00066-t003:** Symptoms that correlate with the severity of fatigue according to the composite symptoms scores from the DSQ-2 questionnaire.

Question from DSQ-2, (Number of Question in DSQ-2)	Symptom’s Domain	r; *p* Value
Dead, heavy feeling after starting to exercise (14) ^1^	PEM	0.62; <0.0001
Next-day soreness or fatigue after non-strenuous, everyday activities (15)	PEM	0.63; <0.0001
Mentally tired after the slightest effort (16)	PEM	0.31; 0.02
Minimum exercise makes you physically tired (17)	PEM	0.60; <0.0001
Physically drained or sick after mild activity (18)	PEM	0.71; <0.0001
Feeling unrefreshed after you wake up in the morning (19)	Sleep disorder	0.47; 0.0003
Needing to nap daily (20)	Sleep disorder	0.29; 0.03
Sleeping all day and staying awake all night (24)	Sleep disorder	0.37; 0.01
Pain or aching in your muscles (25)	Pain syndromes	0.38; 0.004
Muscle weakness (33)	Neurological sensory/motor symptoms	0.44; 0.0007
Nausea (48)	Dysautonomia	0.31; 0.02
Feeling unsteady on your feet, like you might fall (49)	Neurological sensory/motor symptoms	0.26; 0.05
Sore throat (64)	Immune dysfunction	0.28; 0.04
Tender/Sore lymph nodes (65)	Immune dysfunction	0.27; 0.045
Flu-like symptoms ^2^ (67)	Immune dysfunction	0.30; 0.02
Muscle fatigue after mild physical activity (75)	PEM	0.31; 0.02
Worsening of symptoms after mild mental activity (77)	PEM	0.32; 0.01
Daytime drowsiness (84)	Sleep disorder	0.32; 0.02
Sinus infections ^3^ (87)	Immune dysfunction	0.27; 0.04
Urinary urgency (88)	Dysautonomia	0.39; 0.01

^1^ (Number) represent the number of question in the DSQ-2 questionnaire. ^2^ A combination of several of the following symptoms: high body temperature, headaches and muscle aches, cough, sore throat, severe fatigue, nasal congestion or runny nose, chills, nausea, or vomiting. ^3^ Symptoms of sinusitis (discomfort in the cheekbones, bridge of the nose, or above the eyes, often accompanied by persistent headache, nasal congestion, persistent nasal discharge).

**Table 4 diagnostics-13-00066-t004:** Medians of the severity of depressive symptoms in the cohorts according to HADS-D subscale scores, r—effect size.

	1st Cohort (ME/CFS)	2nd Cohort (Post-COVID ME/CFS)	3rd Cohort (Healthy Controls)	*p* Value, (r)	
				1st vs. 3rd	1st vs. 3rd
Severity of depressive symptoms, HADS-D score	8.0 [6.0; 11.0]	8.5 [4.75; 12.0]	3.0 [1.0; 8.0]	0.01. (0.120)	0.01. (0.120)

**Table 5 diagnostics-13-00066-t005:** Medians of the severity of anxiety symptoms in the cohorts according to HADS-A subscale scores, r—effect size.

	1st Cohort (ME/CFS)	2nd Cohort (Post-COVID ME/CFS)	3rd Cohort (Healthy Controls)	*p* Value, (r)	
				1st vs. 3rd	2nd vs. 3rd
Severity of anxiety symptoms, HADS-A score	8.0 [5.8; 10.0]	9.5 [5.3; 15.3]	4.0 [1.0; 8.5]	0.04, (0.088)	0.06, (0.216)

**Table 6 diagnostics-13-00066-t006:** Results of the correlation analysis between the severity of the depressive or anxiety symptoms and the severity of fatigue.

		r; *p*
Severity of fatigue (the composite score of the 13th question of the DSQ-2 questionnaire)	Severity of depressive symptoms (HADS-D score)	0.11; 0.44
Severity of fatigue (the composite score of the 13th question of the DSQ-2 questionnaire)	Severity of anxiety (HADS-A score)	−0.18; 0.22

**Table 7 diagnostics-13-00066-t007:** Prevalence of POTS in the participants with ME/CFS developed outside the context of COVID-19, ME/CFS that developed after COVID-19, and healthy controls, OR—odds ratio.

	1st Cohort (ME/ CFS)	2nd Cohort (Post-COVID ME/CFS)	3rd Cohort (Healthy Controls)	*p* Value, (OR)	
				1st vs. 3rd	2nd vs. 3rd
Prevalence of POTS	6 (37.5%)	6 (75.0%)	1 (11.1%)	0.35	0.02, (24.00)

**Table 8 diagnostics-13-00066-t008:** Characteristics of microcirculation assessed with laser Doppler flowmetry in the participants with ME/CFS developed outside the context of COVID-19, in the participants with ME/CFS that developed after COVID-19, and in healthy controls. M—average value of tissue perfusion with blood, σ—mean square deviation of M oscillations in a given time interval, vALF—contribution of low-frequency oscillations (0.05–0.2 Hz) to the total power of the spectrum of biorhythms; vAHF—contribution of high-frequency oscillations (0.2–0.4 Hz) to the total power of the spectrum of biorhythms; vACF—contribution of pulse frequency oscillations (0.8–1.6 Hz) to the total power of the spectrum of biorhythms; IFM—the oscillation index; R—vascular resistance; CT—microvascular tone, r—effect size.

Parameters	1st Cohort (ME/CFS)	2nd Cohort (Post-COVID ME/CFS)	3rd Cohort (Healthy Controls)	*p* Value, (r)		
				1st vs. 3rd	2nd vs. 3rd	2nd vs. 3rd
M, perfusion units	4.86 [4.55; 6.88]	5.84 [4.13; 6.74]	3.08 [2.19; 4.71]	0.02, (0.432)	0.01, (0.655)	0.72
σ, perfusion units	0.59 [0.46; 0.91]	0.70 [0.51; 0.83]	0.69 [0.52; 0.95]	0.65	0.95	0.52
vALF, perfusion units	16.96 [8.43; 24.36]	15.94 [10.17; 21.57]	38.08 [22.28; 43.82]	0.05, (0.318)	0.02, (0.580)	0.98
vAHF, perfusion units	16.97 [5.42; 27.31]	4.11 [3.35; 29.88]	5.42 [2.81; 14.65]	0.25	0.94	0.32
vACF, perfusion units	68.09 [48.26; 76.42]	76.52 [59.96; 81.22]	54.87 [45.20; 67.23]	0.35	0.30	0.35
IFM,	0.33 [0.23; 0.47]	0.36 [0.24; 0.42]	0.60 [0.41; 0.70]	0.04, (0.354)	0.04, (0.510)	0.94
R	1.01 [0.86; 1.16]	0.82 [0.66; 0.96]	0.83 [0.70; 0.92]	0.09	0.94	0.04 (0.350)
CT	2.77 [2.28; 3.71]	3.32 [3.00; 6.20]	2.77 [2.14; 3.80]	0.94	0.18	0.15

## Data Availability

Not applicable.

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
