# Peer review of "Myalgic Encephalomyelitis/Chronic Fatigue Syndrome and Post-COVID Syndrome: A Common Neuroimmune Ground?"

_diagnostics, 2022, doi:10.3390/diagnostics13010066_

Round 1

Reviewer 1 Report

Introduction:

Introduction needs to be ended with hypotheses that were tested using statistical analysis in the current study

Methods:

Could You provide data on how many patients diagnosed as ME/CFS fulfilled each criteria? And what was the overlap between those subgroups?

Why 75 years old was the upper range? More information on the age range of participants should be provided in the results section.\

The normality of continuous variables 189 distribution was evaluated with the Shapiro–Wilk test” Please note that You should also visually inspect histograms

“non-normal variables” please note that You should refer to the distribution not variables per Se

You should provide information on post-hoc tests applied.

Results

Could provide the effect size for significant comparisons?

More extensive description of table 2, what is put inside square brackets?

Is that a confidence interval?

I believe that at least some results, as in the Table 10 would more clear if presented on figures with significant results of post-hoc testing provided.

Discussion

In general I do enjoy figures in discussion. However, data gathered and

results of analysis are not an enough evidence on what figures represent.

Therefore, I would describe figures as description of a hypothesis generated based on the current study conclusions. However, please note that  those hypotheses need to be confirmed in the further studies.

Regarding Figure 2: are You sure that in the top panel PEM should be connected to sleep disorder only and not to the “pain syndromes”? Are those entities related to just up to two others? In my opinion, most of those circles should ne interconnected. In my humble opinion such hypothesis could be derived from the so far existing literature, as:

“Network Analysis of Symptoms Co-Occurrence in Chronic Fatigue Syndrome. Int J Environ Res Public Health. 2021. PMID: 34682478”

“Post-Exertional Malaise Is Associated with Hypermetabolism, Hypoacetylation and Purine Metabolism Deregulation in ME/CFS Cases” Diagnostics (Basel). 2019 Jul 4;9(3):70. doi: 10.3390/diagnostics9030070.”

“Pain-Related Post-Exertional Malaise in Myalgic Encephalomyelitis / Chronic Fatigue Syndrome (ME/CFS) and Fibromyalgia: A Systematic Review and Three-Level Meta-Analysis: Pain Med. 2022 May 30;23(6):1144-1157. doi: 10.1093/pm/pnab308.

Author Response

Response to Reviewer 1 Comments

Thank you for your letter and for the opportunity to revise our paper “Myalgic Encephalomyelitis/Chronic Fatigue Syndrome and Post-COVID Syndrome: A Common Neuroimmune Ground?”. The suggestions offered by both reviewers have been immensely helpful, and we also appreciate your insightful comments.

Point 1:  Introduction needs to be ended with hypotheses that were tested using statistical analysis in the current study

Response 1: We agree with the Reviewer and included the hypothesis in the end of the Introduction

Point 2: Could You provide data on how many patients diagnosed as ME/CFS fulfilled each criteria? And what was the overlap between those subgroups?

Response 2: We added the data, requested by the Reviewer in the section 3.1, and illustrated the overlap in Figure 1.  

Point 3: Why 75 years old was the upper range? More information on the age range of participants should be provided in the results section.

Response 3: We decided to establish 75 years old as the upper range in inclusion criteria since the prevalence of microcirculation disorders related to the comorbid cardiovascular conditions grow with the age of the patient.  The comorbid conditions could hamper the interpretation of the results. Besides that, the prevalence of ME\CFS in individuals older then 75 years old is lower, and fatigue is rather caused by another reasons. We agree with the Reviewer and provided data on the median age of participant (with interquartile range) in the results section.

Point 4: “The normality of continuous variables distribution was evaluated with the Shapiro–Wilk test” Please note that You should also visually inspect histograms

Response 4: We agree with the Reviewer and hasten to assure the Reviewer that the histograms were also visually inspected. We replaced the mentioned sentence with “The data normality was assessed using the ShapiroWilk test and visual histogram inspection”

Point 5: “non-normal variables” please note that You should refer to the distribution not variables per Se

Response 5: We agree with the Reviewer and replaced “non-normal variables” with “non-normal distribution

Point 6: You should provide information on post-hoc tests applied.

Response 6: We agree with the Reviewer and provide the requested information in 2.3 section (“When results were significant, they were further explored using the post-hoc Dunn’s test”).

Point 7: Could provide the effect size for significant comparisons?

Response 7:  We are grateful to the Reviewer for this comment. We included the information of the effect size in 2.3 section and provided the effect size for significant comparisons in Results section.  

Point 8: More extensive description of table 2, what is put inside square brackets? Is that a confidence interval?

Response 8: We are grateful to the Reviewer for pointing this out. Interquartile range is put inside square brackets, and we indicated this in the table 2.

Point 9: I believe that at least some results, as in the Table 10 would clearer if presented on figures with significant results of post-hoc testing provided.

Response 9 We are grateful to the Reviewer for this insightful comment and presented the results from table 9 (significant after post-hoc) in Figure 2.

Point 10: I would describe figures as description of a hypothesis generated based on the current study conclusions. However, please note that those hypotheses need to be confirmed in the further studies.

Response 10: We agree with the Reviewer and add in the end of the Discussion that the hypotheses which are reflected in the Figures need to be confirmed in the further studies.

Point 11Regarding Figure 2: are You sure that in the top panel PEM should be connected to sleep disorder only and not to the “pain syndromes”? Are those entities related to just up to two others? In my opinion, most of those circles should be interconnected. 

Response 11: We completely agree with the Reviewer! We did not intend to show the connection between these symptom cluster in this Figure, but rather their relation to chronic fatigue. In order to make it clearer, we removed the connection between the circles in the top line.

Reviewer 2 Report

The article by Ryabkova et al titled “ Myalgic Encephalomyelitis/Chronic Fatigue Syndrome and 2 Post-COVID Syndrome: A Common Neuroimmune Ground ” tried to investigate whether the myalgic Myalgic Encephalomyelitis/Chronic Fatigue Syndrome and 2 Post-COVID Syndrome have a shared core.

The article is highly confused and not well structured, the author needs to re-write it and make it clearer, below are some points should to be addressed:

1- the introduction is not sufficiently expressed the topic of the article

2- patients were recruited in this study are self-diagnosed and this is I think not valid, the clinician should diagnose patients according to the international standard.

3- results section are too long and highly confused, the author should remove not important findings.

4- discussion section is too long and general and the author did not discuss their major findings

5- conclusions section should be short as much as the authors can and they should only show what did they conclude based on the aim of the study.

Author Response

Response to Reviewer 2 Comments

Thank you for your letter and for the opportunity to revise our paper “Myalgic Encephalomyelitis/Chronic Fatigue Syndrome and Post-COVID Syndrome: A Common Neuroimmune Ground?”. The suggestions offered by both reviewers have been immensely helpful, and we also appreciate your insightful comments.

Point 1: the introduction is not sufficiently expressed the topic of the article

Response 1: We agree with the Reviewer and re-write the Introduction. We described the current state of the research field regarding neuroimmune ground of ME/CFS and post-COVID-19 syndrome as well as the role of mental health issues. We also formulated the hypothesis which was tested in our study.

Point 2: patients were recruited in this study are self-diagnosed and this is I think not valid, the clinician should diagnose patients according to the international standard.

Response 2: Apparently, we didn't describe the process of the recruitment of the patients clearly enough.  The patients suggested that they suffer from ME/CFS were diagnosed by clinicians according to the international standard (Bateman, 2021). While over 20 case definitions exist for the diagnosis of ME and CFS (Brurberg et al., 2014), we used four the most common case definitions to confirm ME/CFS in our study. In order to avoid subjectivity, DePaul Symptoms Questionnaire-2 was applied. We add the subsection “Patient recruitment” in the manuscript and incorporated the folloing sentence: “After filling out the questionnaires, patients were assessed by clinicians and diagnosed with ME/CFS according to the international standard [Bateman, 2021]”

Brurberg KG, Fønhus MS, Larun L, Flottorp S, & Malterud K (2014). Case definitions for chronic fatigue syndrome/myalgic encephalomyelitis (CFS/ME): A systematic review. BMJ Open, 4(2). doi: 10.1136/bmjopen-2013-003973

Bateman L, Bested AC, Bonilla HF, Chheda BV, Chu L, Curtin JM, Dempsey TT, Dimmock ME, Dowell TG, Felsenstein D, Kaufman DL, Klimas NG, Komaroff AL, Lapp CW, Levine SM, Montoya JG, Natelson BH, Peterson DL, Podell RN, Rey IR, Ruhoy IS, Vera-Nunez MA, Yellman BP. Myalgic Encephalomyelitis/Chronic Fatigue Syndrome: Essentials of Diagnosis and Management. Mayo Clin Proc. 2021 Nov;96(11):2861-2878. doi: 10.1016/j.mayocp.2021.07.004.

Point 3: results section are too long and highly confused, the author should remove not important findings.

Response 3: We are grateful to the Reviewer for this comment and removed not important things.

Point 4: discussion section is too long and general and the author did not discuss their major findings

Response 4: We agree with the Reviewer that discussion section is quite long, but this was necessary to discuss all major findings of the study (according to the tested hypothesis which was defined in the Introduction). The discussion section is structured, divided into paragraphs, and all major findings were discussed (the prevalence of ME/CFS in post-COVID-19; the correlation of the fatigue with other symptom clusters; no association of the fatigue with depression/anxiety; evidence for the immune dysfunction, orthostatic intolerance and microcirculation disorders in ME/CFS and post-COVID-19)

 Point 5: conclusions section should be short as much as the authors can and they should only show what did they conclude based on the aim of the study.

Response 5: We agree with the Reviewer and excluded from the conclusion section the data, which did not relate to the hypothesis of the study, defined in the Introduction

According to the suggestion of the Reviewer, the extensive English language and style revision was also performed

Round 2

Reviewer 2 Report

The authors have addressed all points raised by me